# Facile construction of fully sp$^2$-carbon conjugated two-dimensional covalent organic frameworks containing benzobisthiazole units

Yuancheng Wang [1], Wenbo Hao[1], Hui Liu[1], Renzeng Chen[1], Qingyan Pan[1], Zhibo Li [1✉] & Yingjie Zhao [1✉]

Developing a facile strategy for the construction of vinylene-linked fully $\pi$-conjugated covalent organic frameworks (COFs) remains a huge challenge. Here, a versatile condition of Knoevenagel polycondensation for constructing vinylene-linked 2D COFs was explored. Three new examples of vinylene-linked 2D COFs (**BTH-1**, **2**, **3**) containing benzobisthiazoles units as functional groups were successfully prepared under this versatile and mild condition. The electron-deficient benzobisthiazole units and cyano-vinylene linkages were both integrated into the $\pi$ conjugated COFs skeleton and acted as acceptor moieties. Interestingly, we found the construction of a highly ordered and conjugated D-A system is favorable for photocatalytic activity. **BTH-3** with benzotrithiophene as the donor with a strong D-A effect exhibited an attractive photocatalytic HER of 15.1 mmol h$^{-1}$g$^{-1}$ under visible light irradiation.

[1] College of Polymer Science and Engineering, Qingdao University of Science and Technology, Qingdao 266042, China. ✉email: zbli@qust.edu.cn; yz@qust.edu.cn

Covalent organic frameworks (COFs), in which the building blocks were connected covalently and arranged precisely, are attracting increasing attention since the first report by Yaghi's group in 2005[1–11]. The unique features of COFs such as the well-defined structure, high specific surface area, ordered pore channels, etc. endow them with potential applications in gas storage and separation[12–15], catalysis[16–19], sensor[20–22], energy storage[23,24], and optoelectronics[25], etc. So far, the construction of highly ordered structures of COFs mainly relies on the reversibility of dynamic covalent bonds between the building blocks. Schiff base reaction and boronic acid dehydration reaction are frequently used and prone to achieve high crystallinity owing to their dynamic reversibility[2,3]. Although the reversibility of these dynamic covalent bonds endows the COFs good crystallinity, it weakens the π-conjugation and chemical stability of these materials[26–28]. On one hand, hydrolysis of these dynamic covalent bonds usually occurs under strongly acidic conditions. On the other hand, the poor electron delocalization caused by the strongly polarized nature of the dynamic covalent bonds such as the imine bond will hamper the charge carrier mobility of the COFs[26,28]. Developing COFs with fully π-conjugated structures and excellent chemical stability is highly desirable. Enormous efforts have been done to explore the facile access to highly stable and fully π-conjugated COFs[26–28]. New chemistry reactions for constructing conjugated COFs with optimum electron delocalization have been explored. Oxazole[18,29–31], thiazole[31–33], vinylene[34–42], pyrazine[43,44], imidazole[45], and thieno[3,2-c]pyridine[46] as stable and conjugated linkages have been developed recently. These π-conjugated COFs have displayed exceptional properties and greatly broadened the application fields. However, the study of these fully π-conjugated COFs is still in its infancy stage either in structural diversities or applications. The synthesis of these non-imine COFs is usually tricky with a low success rate and bad generality. This is understandable in consideration of the bad reversibility of the coupling reactions. To simultaneously gain desirable conjugation and refined crystallinity within the COF structure is still highly challenging.

Knoevenagel polycondensation has been successfully used to construct fully π-conjugated vinylene-linked 2D COFs[40,47]. Benefiting from the extended π conjugation along with the 2D skeletons, these cyano-vinylene-linked 2D COFs have shown excellent performance in magnetic properties[40,48], photoluminescence[49,50], photocatalysis[35,51–54], and energy storage[55]. Herein, we developed a facile way to prepare the cyano-vinylene-linked 2D COFs through Knoevenagel polycondensation. A new reaction mechanism was proposed. The neutral AcONH$_4$ was used to catalyze the condensation, which was supposed to enhance the reversibility of the initial step of Knoevenagel polycondensation. This strategy shows better success rates and universality. Three new fully sp$^2$-carbon π-conjugated COFs containing benzobisthiazole (BTH) as functional groups were constructed by Knoevenagel polycondensation under this new explored reaction condition. The integration of BTH unit and cyano-vinylene linkage provides an electron-deficient π-conjugated C2 symmetrical building block (Fig. 1). Other functional moieties such as triazine and benzotrithiophene were introduced as the C3 symmetrical units. The semiconductor properties of the π-conjugated COFs could be tuned via altering the electronic properties of C3 symmetrical units. Thanks to the extended π-electron conjugation and highly ordered D-A structure, BTH-3 with benzotrithiophene as electron-donor unit exhibited an attractive photocatalytic HER of 15.1 mmol h$^{-1}$ g$^{-1}$ under visible light irradiation.

## Results

### Synthesis and characterization of benzobisthiazole-based COFs.
Compound BTH-2CN was prepared according to the reported literature[56]. The electron-withdrawing effect of benzobisthiazole and cyano units increased the activity of the CH$_2$ protons. Up to now, the cyano-vinylene-linked 2D COFs were mainly prepared via base-catalyzed Knoevenagel condensation condition[57]. We first selected TA as the C3 symmetrical aldehyde and screened the base catalysts including DBU, Cs$_2$CO$_3$, and NaOH. Unfortunately, no sign of crystallinity was observed under these base-catalyzed conditions. During the condensation reaction, the formation of carbanion was the initial step of condensation (Fig. 1a). The electron-deficient BTH unit and the cyano group were supposed to stabilize the carbanion intermediate, which led to the poor reversibility of the first step. Ammonium acetate was then adopted for the catalyst system. Different from the alkaline conditions, the Schiff base formation with excellent reversibility was supposed to become the initial step (Fig. 1b)[58]. A model reaction between BTH-2CN and 4-tert-butylbenzaldehyde was first performed to explore the feasibility (Supplementary Fig. 1). The reaction was further used to prepare conjugated COFs. Fortunately, crystalline BTH-1 was finally obtained in a mixture of tetrahydrofuran/ ammonium acetate under 100 °C for 3 days. To explore the adaptability of this reaction condition and diversify the structures to adjust the semiconductor properties of the COFs in the meanwhile, two more C3 symmetrical monomers of TP and TS were selected to construct a series of fully sp$^2$-carbon conjugated COFs, respectively (Fig. 1c). BTH-1 and 2 were both obtained as yellow powders, while BTH-3 was obtained as a deep red powder. Similar to most of the reported COFs, these conjugated COFs are also insoluble in common solvents.

The $^{13}$C CP-MAS NMR and FT-IR spectra were first performed to verify the chemical structures and components. As displayed in Fig. 2, the peaks appearing at 170 ppm confirmed the existence of the thiazole ring. The peaks at about 110 and 120 ppm further proved the formation of vinylene linkage and the existence of cyano units. These distinguishing features indicated the successful condensation of the monomers. FT-IR spectra of the COFs, monomers and the model compound also provided strong evidence for the structural identification (Supplementary Fig. 2). The stretching modes characteristic of the cyano group were observed at about 2216 cm$^{-1}$ for all the three COFs. The signals at about 1553 cm$^{-1}$ could be attributed to the C=N bond stretching of thiazole rings. All the results above suggest the successful preparation of BTH-1, 2, 3. The thermal stabilities of the three COFs were then evaluated via thermogravimetric analysis (Supplementary Fig. 3). All of them displayed good stability with decomposition temperature up to 300 °C. The morphologies of BTH-1, 2, 3 were further explored by scanning electron microscopy (SEM) and transmission electron microscopy (TEM) (Supplementary Fig. 4). The SEM images of BTH-1, 3 displayed block-like particles, while a fibrous morphology was observed in BTH-2. Although the morphologies were different, obvious layered structures could be observed for all the three BTH COFs by TEM images.

Powder X-ray diffraction was performed to assess the crystallinities of BTH-1, 2, 3. The detailed structure information was given by comparison of the experimental PXRD profiles and the predicted profiles. As shown in Fig. 3, the calculated PXRD profiles for the AA stacking mode were found to match excellently with the experimental profiles of BTH-1, 2, 3, respectively. The optimized PXRD data suggested that the (100) crystal plane of BTH-1, 2 appeared at a similar position of $2\theta = 2.3°$. Pawley refinement of the AA stacking model based on the experimental profile gave unit cell with parameters ($a = b = 45.07$ Å, $c = 3.49$ Å, $\alpha = \beta = 90°$, $\gamma = 120°$) for BTH-1 (residuals $R_p = 5.69\%$, $R_{wp} = 7.01\%$) and ($a = b = 45.58$ Å, $c = 3.48$ Å, $\alpha = \beta = 90°$, $\gamma = 120°$) for BTH-2 (residuals $R_p = 8.16\%$, $R_{wp} = 6.64\%$), consistent with hexagonal unit cell. The optimized

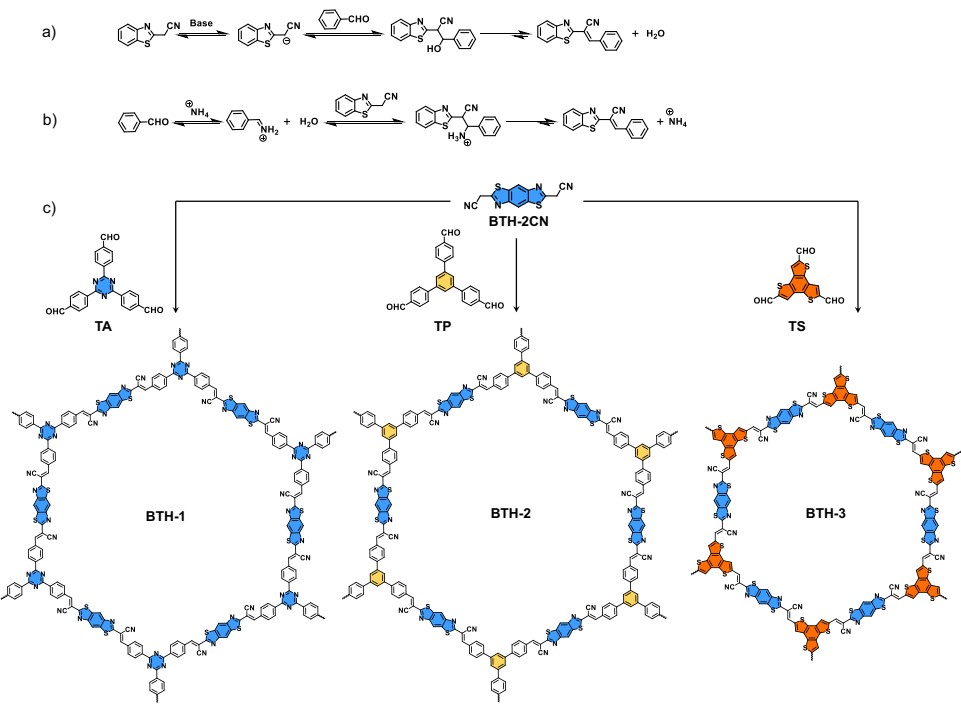

**Fig. 1 Chemical structures. a, b** Proposed mechanisms of Knoevenagel polycondensation at different conditions. **c** Synthetic routes and chemical structures of COFs **BTH-1**, **2**, **3**.

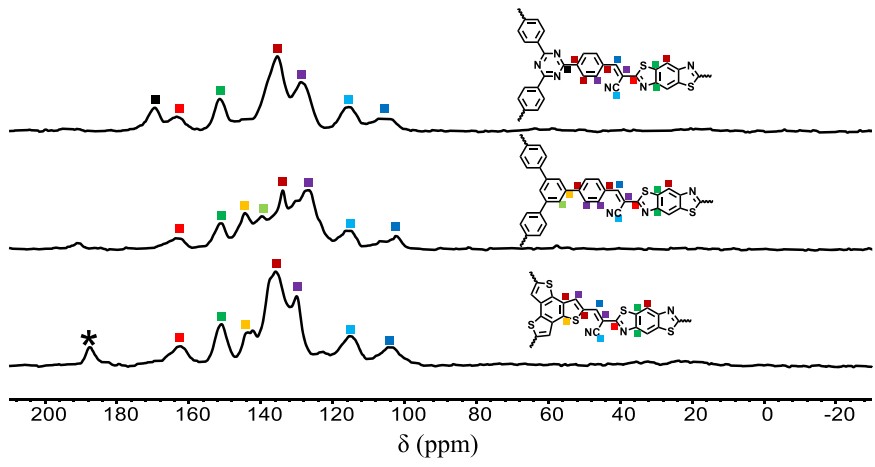

**Fig. 2 Solid-state $^{13}$C CP-MAS NMR spectra of BTH-1, 2, 3.** * signal of residual aldehyde group.

PXRD of **BTH-3** displayed a (100) reflection at $2\theta = 2.8°$ due to the smaller pore size. Pawley refinement data exhibited cell parameters of $a = b = 37.45$ Å, $c = 3.52$ Å, $\alpha = \beta = 90°$, $\gamma = 120°$, revealing minimal difference ($R_p = 4.09\%$ and $R_{wp} = 5.57\%$). Furthermore, the porosities were further examined via nitrogen adsorption experiments at 77 K. The sharp rise in the low-pressure range, as a characteristic of the type I sorption isotherm, indicate their microporous characteristics. The Brunauer–Emmett–Teller (BET) surface areas were observed to be 644, 686, 1140 m$^2$ g$^{-1}$ for **BTH-1**, **2**, **3**, respectively. The pore size distribution curves clearly show two different pore limiting diameters (PLD) for **BTH-1**, **2**, **3** (Supplementary Fig. 5). A similar phenomenon has also been observed in the previous report[34,35,37]. Due to the limitation of sp$^2$-carbon double bond formation reaction, stacking faults are indeed expected to broadly exist in the synthesized **COFs**. The interlayer slip or stagger of AA stacking mode induced the diverse pore size distributions.

**Photocatalytic and optoelectronic properties.** Considering the unique fully conjugated 2D structures and the porosities of the obtained COFs, the photocatalytic hydrogen evolution experiments were then performed by suspending the **BTH**-based COFs materials in 0.1 M ascorbic acid solution under the irradiation of a 300 W Xe lamp. The ascorbic acid acted as the optimal sacrificial electron donor. As displayed in Fig. 4a, all the three COFs exhibited photoactivity toward hydrogen evolution, especially for **BTH-1** and **BTH-3** (Supplementary Table 1). **BTH-1** displayed a hydrogen evolution rate (HER) of 10.5 mmol h$^{-1}$ g$^{-1}$ under visible light ($\lambda > 420$ nm), while **BTH-3** reached a higher HER of 15.1 mmol h$^{-1}$ g$^{-1}$. Although **BTH-1, 2** possess similar skeleton structures, **BTH-2** only exhibited an HER of 1.2 mmol h$^{-1}$g$^{-1}$. The apparent quantum yields (AQYs) of **BTH-1** (5 mg) were measured to be 1.569%, 1.089%, 1.925% and 0.75% at 420, 450, 500, and 550 nm, respectively. The narrow bandgap of **BTH-3** endows its

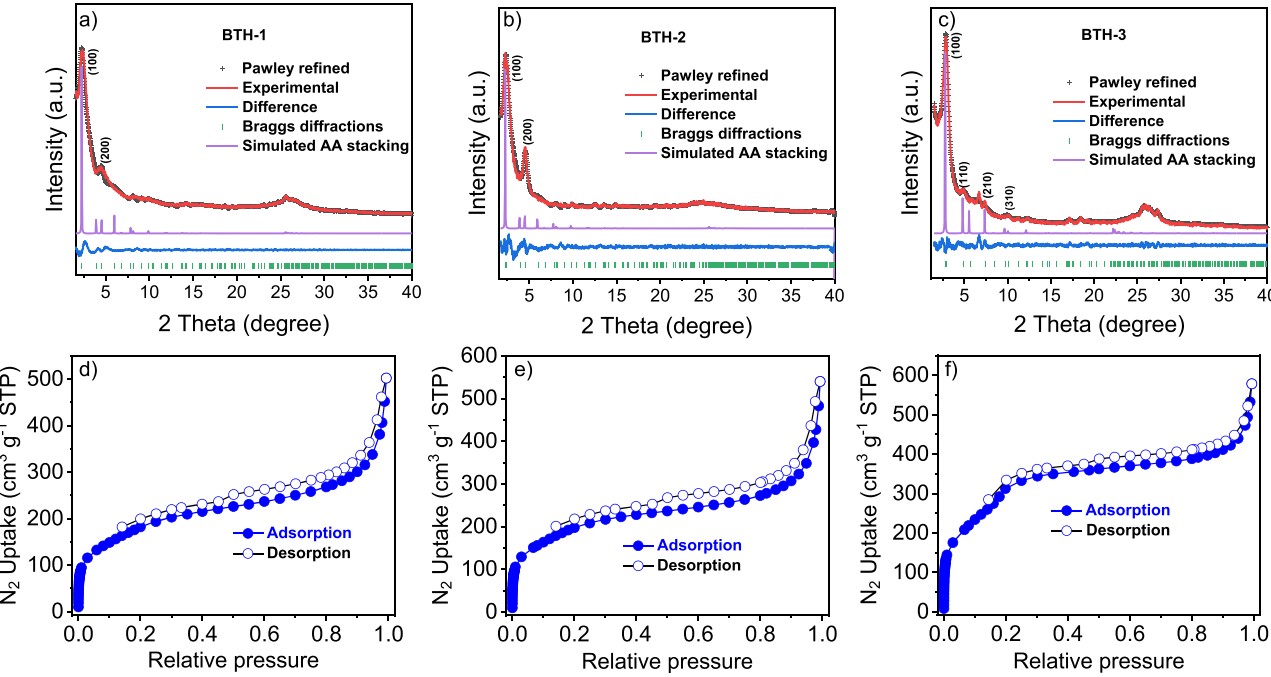

**Fig. 3 Characterizations of the structures. a–c** PXRD patterns of **BTH-1, 2, 3**: experimental pattern (red), Pawley refined profile curve (black), Bragg diffractions (green), difference (blue), simulated patterns from AA eclipsed (purple); nitrogen adsorption and desorption isotherms of **BTH-1** (**d**), **BTH-2** (**e**), and **BTH-3** (**f**).

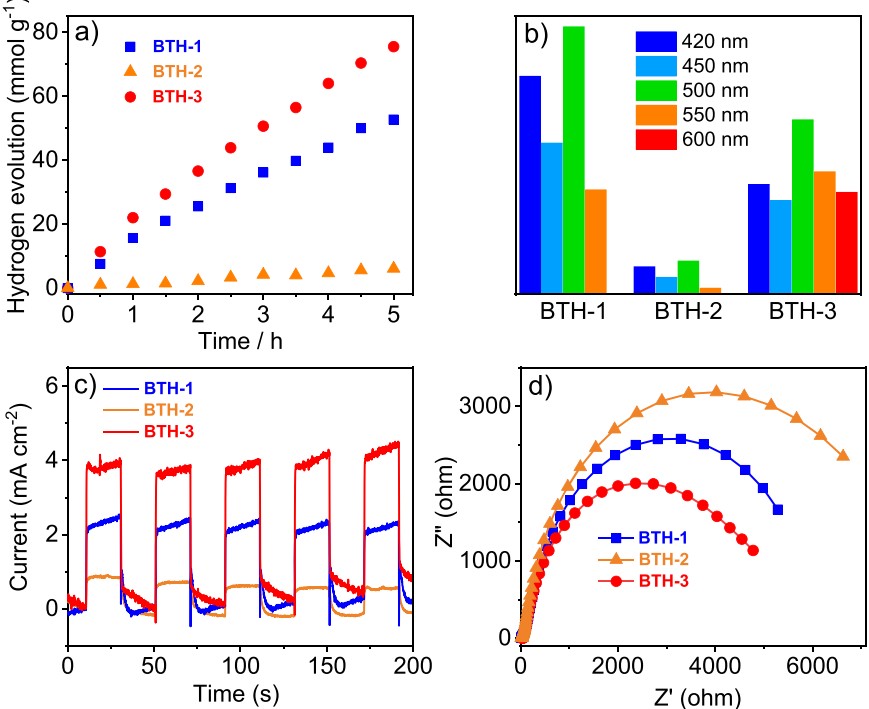

**Fig. 4 Photocatalytic H$_2$ evolution. a** Time course of photocatalytic hydrogen evolution under visible light irradiation (>420 nm) for **BTH-1, 2, 3**. **b** AQY of **BTH-1, 2, 3** under different monochromatic light irradiation. **c** Periodic on/off photocurrent output of **BTH-1, 2, 3** casted on FTO glass. **d** Electrochemical impedance spectroscopy Nyquist plots of **BTH-1, 2, 3**.

good photoactivity even at 600 nm. The AQYs of **BTH-3** is 0.792%, 0.676%, 1.256%, 0.883% and 0.735% at 420, 450, 500, 550, and 600 nm, respectively. The stability test result suggests that **BTH-3** could keep relatively good photocatalytic performance under irradiation (Supplementary Fig. 7). To exclude

the interference factors of the possible impurities, the photocatalytic performance of the monomers and model compound has been further explored. The monomers and model compound did not show any photocatalytic activities towards hydrogen evolution. Besides, control experiments without the

existence of Pt or sacrificial electron donors were also performed. Again no hydrogen evolution process was observed.

To better understand the structure-photocatalytic performance relationship and delve further into the potential mechanism, solid-state absorption spectra were first measured (Supplementary Fig. 6). The absorptions of **BTH-1**, **2**, **3** cover almost the entire visible light region even up to 1000 nm. As a result, the three conjugated COFs are capable of harvesting visible light for photocatalysis efficiently. Owing to the strong intramolecular donor-acceptor effect in **BTH-3**, a strong absorption from 400 to 800 nm which overlaps with the solar energy region was observed. The bandgaps of **BTH-1**, **2**, **3** are calculated to be 1.91, 2.02, and 1.42 eV based on the absorption spectra, respectively. All the three bandgaps meet the criteria of $\Delta E > 1.23$ eV for photocatalytic hydrogen evolution[59]. For the triazine-based **BTH-1**, much previous research has proven the excellent photocatalytic performance of triazine-containing COFs or CMPs materials generated from the unique charge transport properties of the triazine units[35,46,60–62]. For **BTH-3**, the better performance could be attributed to the strong charge transfer effect, which may prevent inefficient and spontaneous electron-hole recombination[63,64]. To further probe the origins of the difference in photocatalytic performance, the photocurrent measurements of **BTH-1**, **2**, **3** were performed to assess the separation efficiency of the photogenerated electron-hole pairs. The efficient charge separation is the key factor in preventing electron-hole recombination. As depicted in Fig. 4c, the photocurrent responses were completely followed by the photocatalytic activities. **BTH-3** electrode showed a higher photocurrent response than that of **BTH-1** and **BTH-2** electrodes under visible light. Furthermore, the electrochemical impedance spectroscopy suggests **BTH-3** possessed the best electronic conductivity, while **BTH-2** displayed the worst electronic conductivity (Fig. 4d). These results indicated that the charge separation and transport properties were greatly improved due to the introduction of the benzotrithiophene as donor units incorporated the D-A interactions into the conjugated 2D COFs. As a result, a significant improvement of photocatalytic performance was achieved.

## Discussion

In summary, a new condition of the Knoevenagel reaction was first explored for constructing the sp²-carbon conjugated COFs materials which shows good universality and success rate. Three fully sp²-carbon conjugated 2D COFs containing benzobisthiazole units as an electron-withdrawing group were successfully prepared via Knoevenagel polycondensation under this mild condition. The electron-deficient benzobisthiazoles units and cyano-vinylene linkages were both integrated into the π conjugated COFs skeleton. Further introduction of functional building blocks such as triazine and benzotrithiophene could successfully adjust the physical properties of the COFs for photocatalysis applications. It is noted that **BTH-3** containing electron-riched benzotrithiophene as a donor with a strong D-A effect exhibited an attractive photocatalytic HER of 15.1 mmol h⁻¹ g⁻¹ under visible light irradiation, much higher than that of the triazine-containing **BTH-1** (10.5 mmol h⁻¹ g⁻¹) and the benzene-containing **BTH-2** (1.2 mmol h⁻¹ g⁻¹). This research first provides a facile way for the construction of sp²-carbon conjugated COFs. The conventional intractable preparation of the sp²-carbon conjugated COFs was greatly simplified. Second, the fully π conjugated COFs materials with D-A system show excellent photocatalytic hydrogen evolution performance and will definitely promote the development of COFs in photocatalysis.

## Methods

**Synthesis of BTH-1**. A Pyrex tube was charged with **TA** (20.0 mg, 0.051 mmol), **BTH-2CN** (20.6 mg, 0.076 mmol), AcONH₄ (23.4 mg, 0.304 mmol), tetrahydrofuran (2.0 mL). After being degassed by freeze-pump-thaw technique for three times and then sealed under vacuum, the tube was placed in an oven at 100 °C for 3 d. The resulting precipitate was filtered, washed with tetrahydrofuran and ethanol for 2 d, dried at 120 °C under vacuum for 12 h. The activated **BTH-1** was obtained as a yellow powder insoluble in common organic solvents (23.9 mg, yield 63.9%) (¹³C CP/MAS spectra shown as Fig. 2).

**Synthesis of BTH-2**. A Pyrex tube was charged with **TP** (20.0 mg, 0.051 mmol), **BTH** (20.8 mg, 0.077 mmol), AcONH₄ (23.7 mg, 0.308 mmol), tetrahydrofuran (2.0 mL). After being degassed by freeze-pump-thaw technique for three times and then sealed under vacuum, the tube was placed in an oven at 100 °C for 3 d. The resulting precipitate was filtered, washed with tetrahydrofuran and ethanol for 2 d, dried at 120 °C under vacuum for 12 h. The activated **BTH-2** was obtained as a yellow powder insoluble in common organic solvents (25.7 mg, yield 67.7%) (¹³C CP/MAS spectra shown as Fig. 2).

**Synthesis of BTH-3**. A Pyrex tube was charged with **TS** (16.3 mg, 0.049 mmol), **BTH** (20.0 mg, 0.074 mmol), AcONH₄ (22.8 mg, 0.296 mmol), tetrahydrofuran (2.0 mL). After being degassed by freeze-pump-thaw technique for three times and then sealed under vacuum, the tube was placed in an oven at 100 °C for 3 d. The resulting precipitate was filtered, washed with tetrahydrofuran and ethanol for 2 d, dried at 120 °C under vacuum for 12 h. The activated **BTH-3** was obtained as a dark red powder insoluble in common organic solvents (26.3 mg, yield 78.1%) (¹³C CP/MAS spectra shown as Fig. 2).

## Data availability

All data supporting the findings of this study are available within the article, as well as the Supplementary Information file, or available from the corresponding authors.

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

## Acknowledgements

We thank Dr. Guiyuan Wu from Anhui Normal University for the support of the transient absorption spectroscopy measurement. The work was supported by the National Natural Science Foundation of China (21905150, 31202117); the Natural Science Foundation of Shandong Province (No. ZR2020ZD38).

## Author contributions

Y.W. performed the synthesis and the characterizations of the vinylene-linked COFs, including NMR, PXRD, FT-IR, gas absorption. W.H. helped with the synthesis and photocatalytic experiments. H.L. helped with structural characterization analysis. R.C. helped to synthesize the materials. Q.P. helped to perform the high resolution TEM measurements. Y.Z. and Z.L. supervised the experiment. Y.Z. and Y.W. designed the project and wrote the manuscript with contributions from all the authors.

## Competing interests

The authors declare no competing interests.
