## [Peer Review File · Nature Communications]

REVIEWER COMMENTS

Reviewer #1 (Remarks to the Author):

Covalent organic frameworks (COFs) have attracted much attention because of their fascinating structural features for diverse applications. In particular, vinylene-linked 2D COFs have received great interest in the past few years due to the fully conjugated structures and high chemical stability. In this work, Wang et al reported three new examples of vinylene-linked 2D COFs containing benzobisthiazoles by using the well-established Knoevenagel reaction, and conjugated D-A systems were demonstrated for photocatalytic activity with a hydrogen evolution rate of 15.1 mmol h⁻¹ g⁻¹. However, I do not recommend the publication of this work in Nature Comm due to the lack of novelty in application and deep understanding of the synthesis.

More detail comments are as follows:

1. In page 2, the authors mentioned that “we developed a facile way to prepare the cyano-vinylene linked 2D COFs through Knoevenagel Polycondensation”. In fact, the Knoevenagel polycondensation has been well developed for the synthesis of such 2D COFs back to 2016. The only new in this work is that the authors demonstrated a new linker with benzobisthiazole, which is however quite straightforward based on the literature efforts. Therefore, I donot see the clear novelty regarding the new design and synthesis.
2. In page 2, the authors claimed that “The neutral AcONH₄ was used to catalyze the condensation, which consequently enhanced the reversibility of the initial step of Knoevenagel polycondensation.” However, there is no proof in this work which step is the crucial step, and how the modified Knoevenagel reaction becomes more reversible than the base-catalyzed condition? The conclusion made by the authors is very speculative.
3. In Figure 1. There are peaks around 190 ppm, which could result from aldehyde groups, the authors should give the reason for these signals.
4. In Figure S1, the authors should also provide the FT-IR spectra of monomers. Otherwise, it is difficult to assign the peaks.
5. In Page 5, “Although the morphologies are different, obvious layered structures could be observed for all the three BTH COFs by TEM images”. However, the TEM images d and e in Figure S3 are not qualified to confirm the layered structures. The authors shall provide atomic/molecular-level resolution of TEM characterizations.
6. In Figure 2. The authors assigned many crystal planes in PXRD data. However, the (110), (130), and (230) in Figure a, (110) in Figure b, and (200), (410) in figure c are not obvious. Actually, the intensities of some peaks ranging from 10~20 are comparable with these peaks, which even cannot match with the simulated patterns.
7. There is no any detail for the simulations and refinements for PXRD.
8. The pore size distribution does not fit to the expected chemical structure of 2D COFs. As a result, this work can only use PXRD patterns with un-known side peaks to confirm the structures. Then it is difficult to know whether the following good performance come from the 2D COF structure or impurities (there is also no model reaction).

9. The application for photocatalytic water splitting is not new for such COF materials. There have been many 2D COFs which can give much higher photocurrent density, but the HER rates are lower than this paper (J. Am. Chem. Soc. 2020, 142, 4862-4871; Angew. Chem. Int. Ed. 2018, 57, 11968-11972). As mentioned in last point, the authors need to provide solid explanation for the high hydrogen evolution rate, and only the one sentence in summary "Three fully sp²-carbon conjugated 2D COFs containing benzobisthiazole units as photoactive group" is very superficial. Moreover, the authors need to fairly compare the COF performance for the photocatalytic properties with other porous polymer materials, rather than just COFs.

10. In page 8, the authors mentioned that "The efficient charge separation is the key factor in preventing electron-hole recombination". However, there is no experimental data to confirm the advantage. Again, the conclusion is quite speculative.

Reviewer #2 (Remarks to the Author):

This is a wonderful work by Zhibo Li and Ying Jie Zhao and co-workers on the topic of "Facile construction of fully sp²-carbon conjugated two-dimensional covalent organic frameworks containing benzobisthiazole units". This manuscript has been written very well, and the logic behind this work is very sound. Hence, I recommend publication per minor revision.

1. I think COF literature should be cited a bit more in detail [J. Am. Chem. Soc. 2019, 141, 51, 20371–20379; J. Am. Chem. Soc. 2020, 142, 18, 8252–8261; J. Am. Chem. Soc. 2021, 143, 2, 955–963].

2. Solid-state ¹³C CP-MAS NMR spectra of BTH-1, 2, 3. are very broad in nature., I would recommend collecting those once more and with a longer time.

3. Pawley refinement can not provide a cell parameter with four points after the decimal. May check that aspect.

4. Authors mentioned that "all the three COFs exhibited photoactivity toward hydrogen evolution" can they make a comparison table and state where these materials fit in the profile.

Reviewer #3 (Remarks to the Author):

Vinylene-linked 2D COFs have exhibited exceptional properties owing to their high chemical stability and fully sp² carbon conjugated structures. However, the synthesis of vinylene-linked COFs have proven to be a big challenge, which have greatly limited their applications. In addition, the exploration of their diversified structures is also urgently needed. In this manuscript, Wang et al. developed a versatile and mild condition for preparing vinylene-linked 2D COFs. A novel benzobisthiazole-based building block was proposed to construct vinylene-linked 2D COFs via Knoevenagel polycondensation. The triazine-free BTH-3 with benzotrithiophene as the donor with a strong D-A effect exhibited an attractive photocatalytic HER of 15.1 mmol h⁻¹g⁻¹ under visible light irradiation. This research may provide some useful information to address the key problem of the synthesis of fully sp² carbon conjugated COFs and also promote the development of these materials either in synthesis or applications. I would recommend publication after minor revision.

1. The authors claim BTH-3 exhibits the best photocatalytic activity due to the D-A system of TS as donor and BTH as acceptor in the 2D structure. This is reasonable since the D-A system is usually favorable for photocatalytic activity. However, BTH-1 containing TA and BTH units with the weakest D-A effect also displays high photocatalytic performance. The result appeared to be in conflict with the observation of BTH-3. The author should explain the reason.
2. As depicted in Figure 2, all three COFs have good XRD signals. This reviewer also noticed that in the previous reported (Angew. Chem. Int. Ed. 2019, 58, 5376-5381) by the same group. They observed the honeycomb-like internal structure of TP-COF from HRTEM. I am very curious if the same phenomenon can be observed in this work since the sp² carbon conjugated COFs is normally more robust and tolerant to the electron beam of the HRTEM than imine-based COFs.
3. The organic photocatalysts often suffer photo erosion. So how about the photocatalytic stability of the vinylene-linked COFs? The stability testing of the obtained COFs should be performed.
4. In the hydrogen evolution experiment, the Pt and sacrificial electron donor were used in the system. Some control experiments without the existence of Pt or sacrificial electron donors should be performed.
5. To better understand the mechanism of the photocatalytic process and the structure-photocatalytic performance relationship, the fluorescence spectra and fluorescence lifetime of the obtained sp² carbon conjugated COFs may be performed to explore the separation of photo-induced carriers and radiative recombination of photogenerated excitons.
6. The authors need to explain the reason that the apparent quantum yields (AQYs) at 420 nm and 450 nm are lower than those at 500 nm.
7. As a highlight of this manuscript, the mechanism of synthesis should be further clarified. As depicted in Scheme 1b, how the carbon anion was generated? Was the hydrogen of methylene removed by AcO-?
8. Brunauer-Emmett-Teller (BET) surface of BTH-3 (1140 m² g⁻¹) is much higher than BTH-1 (644 m² g⁻¹) and BTH-2 (686 m² g⁻¹). How do the authors exclude the influence of the better BET surface areas of BTH-3 on photocatalytic performance than BTH-1 and BTH-2?
9. The literature references need to be revised. The following references (Angew. Chem. Int. Ed. 2019, 58, 6430-6434; Chem 2019, 5, 1632-1647; Nat. Commun. 2020, 11, 436; J. Am. Chem. Soc. 2021, 143, 369-381; Nat. Commun. 2021, 12, 1354.) on vinylene-linked 2D COFs and photocatalysis of COFs materials should be added.

Reviewer 1:

Covalent organic frameworks (COFs) have attracted much attention because of their fascinating structural features for diverse applications. In particular, vinylene-linked 2D COFs have received great interest in the past few years due to the fully conjugated structures and high chemical stability. In this work, Wang et al reported three new examples of vinylene-linked 2D COFs containing benzobisthiazoles by using the well-established Knoevenagel reaction, and conjugated D-A systems were demonstrated for photocatalytic activity with a hydrogen evolution rate of 15.1 mmol h⁻¹ g⁻¹. However, I do not recommend the publication of this work in Nature Comm due to the lack of novelty in application and deep understanding of the synthesis.

Response: We thank the reviewer for the comments on this manuscript. The novelty of this work mainly lies in the structures and synthetic strategy of the vinylene-linked 2D COFs. Although Knoevenagel reaction has been used to construct COFs since 2016, the structural diversity remains to be greatly limited owing to the lack of new monomers. On the other hand, it is still difficult to achieve the crystalline COFs under alkaline conditions due to the poor reversibility of Knoevenagel reaction. A huge amount of work is often required to screen the reaction conditions and the results are often disappointing. As a result, vinylene-linked 2D COFs with strong donor-acceptor systems have been rarely reported. In this manuscript, a series of vinylene-linked benzobisthiazole-based COFs were successfully constructed using mild and neutral AcONH₄ as the catalyst. The synthetic strategy displayed good adaptability. Furthermore, benzobisthiazole as an electron-deficient and N, S-codoped unit was firstly incorporated into conjugated COFs. Thus, the synthesis of the fully conjugated benzobisthiazole-based COFs is the most significant part of this manuscript. Seeking novel applications for the conjugated COFs is definitely an important part of paving the development of COFs. However, the development of the reported application also needs continuous research either in new materials or the mechanisms to form regular patterns guiding the design and synthesis of materials. Photocatalytic water splitting

as the most ideal method for hydrogen evolution still stay at the laboratory stage and remains to be explored. Developing new materials and exploring the structure-performance relationship plays an important role. The original intention of this research is not only to realize the high efficiency of HER but also to study the structure-photocatalytic performance relationship and explore the strategy for the design of materials with high HER efficiency.

1. In page 2, the authors mentioned that “we developed a facile way to prepare the cyano-vinylene linked 2D COFs through Knoevenagel Polycondensation”. In fact, the Knoevenagel polycondensation has been well developed for the synthesis of such 2D COFs back to 2016. The only new in this work is that the authors demonstrated a new linker with benzobisthiazole, which is however quite straightforward based on the literature efforts. Therefore, I donot see the clear novelty regarding the new design and synthesis.

Answer: Thanks for the reviewer’s good question. This reviewer is right and very familiar with the development of cyano-vinylene-linked 2D COFs. Although Knoevenagel reaction has been used to construct COFs since 2016, it remains a great challenge to achieve the crystalline structure due to the poor reversibility of this reaction. The construction of these COFs often requires harsh conditions and a large number of failures. A reaction condition with universality for various structures of vinylene-linked COFs is urgently needed. In addition, it is difficult to construct vinylene-linked 2D COFs with new functional moieties due to the lack of new monomers with active methylene. This greatly limits the structural diversity of the vinylene-linked 2D COFs. In this work, a new reaction condition of Knoevenagel reaction (neutral AcONH_4) was adopted to the construction of vinylene-linked 2D COFs which shows nice universality. Three new vinylene-linked 2D COFs were successfully obtained under this new condition of Knoevenagel reaction. More successful examples are on the way. This facile preparation method with strong universality encouraged us to explore more interesting structures with functional units.

Integrating the periodic donor-acceptor structure into fully conjugated COFs is another great challenge, which has been rarely reported. So far, there is only one reference associated with the synthesis and application of BTH-2CN (Angew. Chem. Int. Ed. 2011, 50, 1406-1409). Moreover, the BTH-CN and benzobisthiazole units were used to construct vinylene-linked 2D COFs for the first time. Besides, periodic strong donor-acceptor systems were successfully integrated into the vinylene-linked 2D COFs, in which the benzobisthiazole unit and cyano group act as electron acceptor and benzotrithiophene act as the electron donor. In summary, although Knoevenagel reaction is not new for constructing COFs, the new condition explored here could greatly improve the success rate during the synthesis of vinylene-linked 2D COFs with higher efficiency. This finding may pave a new path for the development of vinylene-linked COFs. More and more interesting structures of the vinylene-linked 2D COFs with functional moieties could be prepared by using this facile method.

2. *In page 2, the authors claimed that “The neutral AcONH₄ was used to catalyze the condensation, which consequently enhanced the reversibility of the initial step of Knoevenagel polycondensation.” However, there is no proof in this work which step is the crucial step, and how the modified Knoevenagel reaction becomes more reversible than the base-catalyzed condition? The conclusion made by the authors is very speculative.*

Answer: Thanks for the reviewer’s question. This is a good question. The Schiff base reaction between the aldehyde and NH₄⁺ will first happen under the existence of AcONH₄. This could refer to the Mannich reaction when using acid condition. The Schiff base reaction has been proved to possess excellent reversibility and extensively used to prepare COFs with high crystallinity. Furthermore, the deamination is more difficult to happen compared to the dehydration, which gives it more time to adjust structures. The conclusion was made based on the theory of organic chemistry and the reported literature (Acc. Mater. Res. 2021, 2, 252-265).

3. *In Figure 1. There are peaks around 190 ppm, which could result from aldehyde*

groups, the authors should give the reason for these signals.

Answer: Thanks for the reviewer's question. Although equivalent aldehyde and active methylene were added to the reaction systems, the incomplete reaction often occurs. Thus, unreacted groups will appear at the edge part of the 2D framework. This phenomenon could also be found in imine-linked COFs (J. Am. Chem. Soc. 2020, 142, 13162-13169; J. Am. Chem. Soc. 2020, 142, 3736-3741; Nat. Commun. 2018, 9, 2998).

4. In Figure S1, the authors should also provide the FT-IR spectra of monomers. Otherwise, it is difficult to assign the peaks

Answer: Thanks for the reviewer's good suggestion. The FT-IR spectra of the monomers have been added to **Figure S1**.

Figure S1. The FT-IR spectra of **BTH-1, 2, 3** and the monomers.

5. In Page 5, “Although the morphologies are different, obvious layered structures could be observed for all the three BTH COFs by TEM images”. However, the TEM images d and e in Figure S3 are not qualified to confirm the layered structures. The authors shall provide atomic/molecular-level resolution of TEM characterizations.

Answer: Thanks for the reviewer's good suggestion. We have added new TEM images of **BTH-1, 2, 3** and the **Figure S3** has been updated. From these updated and additional TEM images, the morphology of layered structures could be observed.

Figure S3 SEM images of **BTH-1** (a), **BTH-2** (b) and **BTH-3** (c) and TEM images of **BTH-1** (d, g), **BTH-2** (e, h) and **BTH-3** (f, i).

6. *In Figure 2. The authors assigned many crystal planes in PXRD data. However, the (110), (130), and (230) in Figure a, (110) in Figure b, and (200), (410) in figure c are not obvious. Actually, the intensities of some peaks ranging from 10~20 are comparable with these peaks, which even cannot match with the simulated patterns.*

Answer: Thanks for the reviewer's question. We really appreciate the reviewer for the careful reading and the rigorous and detailed evaluation to this manuscript. Compared with the traditional imine-based COFs, the PXRD of the fully sp²-carbon conjugated COFs often fails to show the signals of every crystal plane clearly in consideration of the bad reversibility of the bond formation. According to the suggestion of this reviewer, we carefully modify the assignment of many crystal planes. The controversial labels have been removed.

7. *There is no any detail for the simulations and refinements for PXRD.*

Answer: Thanks for the reviewer's question. The detailed information of the simulations and refinements for PXRD has been added to the supporting information in the revised manuscript.

8. *The pore size distribution does not fit to the expected chemical structure of 2D COFs. As a result, this work can only use PXRD patterns with un-known side peaks to confirm the structures. Then it is difficult to know whether the following good performance come from the 2D COF structure or impurities (there is also no model reaction).*

Answer: Thanks for the reviewer's good question. As mentioned in the manuscript, stacking faults are indeed expected to broadly exist in the synthesized COFs. The interlayer slip or stagger of AA stacking mode induced the diverse pore size distributions. This phenomenon has also been observed in the previous report (Angew. Chem. Int. Ed. 2018, 57, 11968-11972; Angew. Chem. Int. Ed. 2019, 58, 13753-13757; Angew. Chem. Int. Ed. 2019, 58, 12065-12069; J. Am. Chem. Soc. 2019, 141, 6848-6852).

For the question of impurities, this is a good suggestion. During the preparation of the COFs sample, the obtained materials have been washed with tetrahydrofuran and ethanol via soxhlet extraction for 2 d to remove the impurities. According to the suggestion of this reviewer, the model reaction was done. A small molecule as a

reference was synthesized to further prove the feasibility of the strategy on one hand. On the other hand, the unactive catalysis of the reference compound verified the real role of the 2D COFs during the catalysis process. The NMR spectra of the model compound have been added as **Figure S7** and **S8**.

Figure S7 ¹H NMR spectrum of the model compound.

Figure S8 ¹³C NMR spectrum of the model compound.

9. *The application for photocatalytic water splitting is not new for such COF materials. There have been many 2D COFs which can give much higher photocurrent density, but the HER rates are lower than this paper (J. Am. Chem. Soc. 2020, 142, 4862-4871; Angew. Chem. Int. Ed. 2018, 57, 11968-11972). As mentioned in last point, the authors need to provide solid explanation for the high hydrogen evolution rate, and only the one sentence in summary “Three fully sp²-carbon conjugated 2D COFs containing benzobisthiazole units as photoactive group” is very superficial. Moreover, the authors need to fairly compare the COF performance for the photocatalytic properties with other porous polymer materials, rather than just COFs.*

Answer: Thanks for the reviewer’s question. We didn’t realize that we had made a serious mistake on the quantitative unit of the y-axis until we see this question. The correct Figure has been updated. The lower photocurrent density than the two reported references could mainly be attributed to the differences in measurement

conditions. The concentration of Na₂SO₄ solution, the quantity of loaded COFs and the applied bias will lead to the difference in photocurrent density. In this manuscript, we only applied a bias of 0.2 V. Besides, the concentration of Na₂SO₄ is 0.2 M. Both of them are different from the reported literature. In this research, the comparison of the photocurrent density of **BTH-1, 2, 3** is more significant for exploring structure-photocatalytic performance.

10. In page 8, the authors mentioned that “The efficient charge separation is the key factor in preventing electron-hole recombination”. However, there is no experimental data to confirm the advantage. Again, the conclusion is quite speculative.

Answer: Thanks for the reviewer’s question. This perspective has been proposed in the reported literature (Angew. Chem. Int. Ed. 2018, 57, 14188-14192). In this manuscript, photocurrent and electrochemical impedance were also both studied to confirm the advantage.

Reviewer 2:

This is a wonderful work by Zhibo Li and Yingjie Zhao and co-workers on the topic of "Facile construction of fully sp²-carbon conjugated two-dimensional covalent organic frameworks containing benzobisthiazole units". This manuscript has been written very well, and the logic behind this work is very sound. Hence, I recommend publication per minor revision.

1. I think COF literature should be cited a bit more in detail [J. Am. Chem. Soc. 2019, 141, 51, 20371-20379; J. Am. Chem. Soc. 2020, 142, 18, 8252-8261; J. Am. Chem. Soc. 2021, 143, 2, 955-963].

Answer: Thanks for the reviewer’s suggestion. The references have been renewed. These three papers have been added as ref 9-11.

2. Solid-state ¹³C CP-MAS NMR spectra of BTH-1, 2, 3. are very broad in nature., I

would recommend collecting those once more and with a longer time

Answer: Thanks for the reviewer's suggestion. To list the solid-state ^{13}C CP-MAS NMR spectra of **BTH-1**, **2**, **3** together along the y-axis, the intensity of the signals was compressed and the x-axis was enlarged. As a result, the NMR spectra seem to be very broad. Actually, the NMR spectra of **BTH-1**, **2**, **3** are much sharper when separately displayed. New solid-state ^{13}C CP-MAS NMR spectra have been presented below.

Figure R1. Solid-state ^{13}C CP-MAS NMR spectra of **BTH-1**.

Figure R2. Solid-state ^{13}C CP-MAS NMR spectra of **BTH-2**.

Figure R3. Solid-state ^{13}C CP-MAS NMR spectra of **BTH-3**.

3. *Pawley refinement can not provide a cell parameter with four points after*

the decimal. May check that aspect.

Answer: Thanks for the reviewer's suggestion. The cell parameters have been revised.

4. Authors mentioned that "all the three COFs exhibited photoactivity toward hydrogen evolution" can they make a comparison table and state where these materials fit in the profile.

Answer: Thanks for the reviewer's suggestion. A table has been added as **Table S1** as the reviewer's suggestion.

Reviewer 3:

Vinylene-linked 2D COFs have exhibited exceptional properties owing to their high chemical stability and fully sp² carbon conjugated structures. However, the synthesis of vinylene-linked COFs have proven to be a big challenge, which have greatly limited their applications. In addition, the exploration of their diversified structures is also urgently needed. In this manuscript, Wang et al. developed a versatile and mild condition for preparing vinylene-linked 2D COFs. A novel benzobisthiazole-based building block was proposed to construct vinylene-linked 2D COFs via Knoevenagel polycondensation. The triazine-free BTH-3 with benzotrithiophene as the donor with a strong D-A effect exhibited an attractive photocatalytic HER of 15.1 mmol h⁻¹g⁻¹ under visible light irradiation. This research may provide some useful information to address the key problem of the synthesis of fully sp² carbon conjugated COFs and also promote the development of these materials either in synthesis or applications. I would recommend publication after minor revision.

1. *The authors claim BTH-3 exhibits the best photocatalytic activity due to the D-A system of TS as donor and BTH as acceptor in the 2D structure. This is reasonable since the D-A system is usually favorable for photocatalytic activity. However, BTH-1 containing TA and BTH units with the weakest D-A effect also displays high photocatalytic performance. The result appeared to be in conflict with the observation of BTH-3. The author should explain the reason.*

Answer: Thanks for the reviewer's question. This is a very good question. The high photocatalytic performance of BTH-1 could be attributed to the triazine unit, which has been proved efficient for improving the photocatalytic performances (ACS Appl. Mater. Interfaces 2020, 12, 46483-46489; J. Am. Chem. Soc. 2020, 142, 13, 5958-5963). The triazine units are prone to be excited and generate holes under irradiation.

2. *As depicted in Figure 2, all three COFs have good XRD signals. This reviewer*

also noticed that in the previous reported (Angew. Chem. Int. Ed. 2019, 58, 5376-5381) by the same group. They observed the honeycomb-like internal structure of TP-COF from HRTEM. I am very curious if the same phenomenon can be observed in this work since the sp^2 carbon conjugated COFs is normally more robust and tolerant to the electron beam of the HRTEM than imine-based COFs.

Answer: Thanks for the reviewer's suggestion. We appreciate the attention of this reviewer to our previously reported work. We highly agree with the comments. And we also tried our best to find some honeycomb-like internal structure for the new reported COFs. Unfortunately, we didn't observe the same phenomenon in these three samples. The electronic properties and sample dispersion may affect the TEM images. It is still very tricky to obtain the honeycomb-like internal structure from HRTEM for COFs materials.

3. The organic photocatalysts often suffer photo erosion. So how about the photocatalytic stability of the vinylene-linked COFs? The stability testing of the obtained COFs should be performed.

Answer: Thanks for the reviewer's suggestion. A cycling test of **BTH-3** has been performed. As depicted in **Figure S6**, the photocatalytic performance of **BTH-3** displayed relatively good stability.

Figure S6 Cycling test of **BTH-3** under irradiation.

4. In the hydrogen evolution experiment, the Pt and sacrificial electron donor were used in the system. Some control experiments without the existence of Pt or sacrificial electron donors should be performed.

Answer: Thanks for the reviewer's suggestion. Control experiments without the existence of Pt or sacrificial electron donors have been performed. No hydrogen evolution process was observed.

5. To better understand the mechanism of the photocatalytic process and the structure-photocatalytic performance relationship, the fluorescence spectra and fluorescence lifetime of the obtained sp² carbon conjugated COFs may be performed to explore the separation of photo-induced carriers and radiative recombination of photogenerated excitons.

Answer: Thanks for the reviewer's question. Good suggestion! The fluorescence spectra of **BTH-1, 2, 3** were measured. As depicted in **Figure R4**, BTH-3 exhibited a weaker emission. In addition, the fluorescence peak displayed an obvious redshift due to the strong D-A effect. The fluorescence peak of **BTH-1, 2** appeared at almost the same location. However, **BTH-2** showed a stronger emission than **BTH-1**. This could prove that the charge carriers in **BTH-1, 3** are more difficult to recombine than those in **BTH-1**. The average fluorescence lifetimes of **BTH-1, 2, 3** were measured to be 1.58 ns, 1.28 ns and 1.47 ns. The subtle differences failed to give a reasonable explanation due to the very weak fluorescence of the **BTH-3**.

Figure R4 The fluorescence spectra of **BTH-1, 2, 3**.

6. *The authors need to explain the reason that the apparent quantum yields (AQYs) at 420 nm and 450 nm are lower than those at 500 nm.*

Answer: Thanks for the reviewer's question. The apparent quantum yield is related to the absorption and the hydrogen evolution rate. This phenomenon may be caused by the stronger absorption at 500 nm than 420 nm and 450 nm.

7. *As a highlight of this manuscript, the mechanism of synthesis should be further clarified. As depicted in Scheme 1b, how the carbon anion was generated? Was the hydrogen of methylene removed by AcO⁻?*

Answer: Thanks for the reviewer's question. The methylene of BTH-2CN is prone to be removed owing to the electron-withdrawing effect of the benzobisthiazole unit and the cyano group. Thus, the hydrogen of methylene could be easily removed by AcO⁻ to form a carbon anion.

8. *Brunauer-Emmett-Teller (BET) surface of BTH-3 (1140 m² g⁻¹) is much higher than BTH-1 (644 m² g⁻¹) and BTH-2 (686 m² g⁻¹). How do the authors exclude the influence of the better BET surface areas of BTH-3 on photocatalytic performance than BTH-1 and BTH-2?*

Answer: Thanks for the reviewer's question. We observed that the BET surface area is not the key for photocatalytic performance. For example, the BET surface of

BTH-2 is higher than that of **BTH-1**, while **BTH-1** exhibited a much higher HER than that of **BTH-1**. Therefore, the structure of the COFs is the key point to influence photocatalytic performance. BET surface is not the key factor in photocatalytic performance.

9. *The literature references need to be revised. The following references (Angew. Chem. Int. Ed. 2019, 58, 6430-6434; Chem 2019, 5, 1632-1647; Nat. Commun. 2020, 11, 436; J. Am. Chem. Soc. 2021, 143, 369-381; Nat. Commun. 2021, 12, 1354.) on vinylene-linked 2D COFs and photocatalysis of COFs materials should be added.*

Answer: Thanks for the reviewer's suggestion. The references have been added as ref 19, 42, 50, 54, 55.

REVIEWER COMMENTS

Reviewer #1 (Remarks to the Author):

The authors have put in efforts and addressed some relevant issues. However, there are still some open questions to be addressed.

1. In page 2, the authors attempt an explanation for “The neutral AcONH₄ was used to catalyze the condensation, which consequently enhanced the reversibility of the initial step of Knoevenagel polycondensation.” However, the explanation is still unclear: (1) There is no evidence to confirm “The Schiff base reaction between the aldehyde and NH₄⁺ will first happen under the existence of AcONH₄.” Knoevenagel condensation can also proceed quickly; (2) The authors should give more details to elaborate on the sentence of “the deamination is more difficult to happen compared to the dehydration, which gives it more time to adjust structures. The conclusion was made based on the theory of organic chemistry and the reported literature.
2. In Page 5 and Figure S1, the authors mention that “The signals at about 1579 cm⁻¹ could be attributed to the C=N bond stretching of thiazole rings”. However, it is a bit difficult to find similar signals at ~1579 cm⁻¹.
3. In Figure 2. Although some assignments for PXRD have been deleted, some unknown peaks are still there. The pore size distributions are still not good. Considering there are also side peaks for aldehyde groups in solid ¹³C-NMR, it is still difficult to know whether the following good performance comes from the 2D COF structure or possible impurities.
4. The authors still do not provide solid explanation for the high hydrogen evolution rate, and only the one sentence in summary “Three fully sp²-carbon conjugated 2D COFs containing benzobisthiazole units as photoactive group” is still superficial. In particular, the newly added information “control experiments without the existence of Pt or sacrificial electron donors were also performed. No hydrogen evolution process was observed.” Also, the results cannot confirm the photoactive nature of benzobisthiazole groups.

Reviewer #3 (Remarks to the Author):

I am satisfied with the revision and would like to recommend the publication after addressing one minor concern: The authors claimed the lower AQYs at 420 nm and 450 nm than that at 500 nm can be ascribed to the difference of absorption intensity. However, the absorption spectra could not prove this speculation. The authors can give a more reasonable explanation here.

According to reviewers' suggestions, we have conducted additional experiments and carefully revised the manuscript with more convincing experimental results and interpretations. We believe that we have significantly improved our manuscript according to the reviewers' suggestions. Our point-by-point responses to the referees' comments are given below.

Reviewer 1:

The authors have put in efforts and addressed some relevant issues. However, there are still some open questions to be addressed.

- 1. In page 2, the authors attempt an explanation for "The neutral AcONH₄ was used to catalyze the condensation, which consequently enhanced the reversibility of the initial step of Knoevenagel polycondensation." However, the explanation is still unclear: (1) There is no evidence to confirm "The Schiff base reaction between the aldehyde and NH₄⁺ will first happen under the existence of AcONH₄." Knoevenagel condensation can also proceed quickly; (2) The authors should give more details to elaborate on the sentence of "the deamination is more difficult to happen compared to the dehydration, which gives it more time to adjust structures. The conclusion was made based on the theory of organic chemistry and the reported literature."*

Answer: Thanks for the reviewer's good question. AcONH₄ has been extensively used as a catalyst in the synthesis of small organic molecules via Knoevenagel reaction. The reaction mechanism that the aldehyde and NH₄⁺ will first happen under the existence of AcONH₄ has been proposed in the reported research (Tetrahedron Letters 2006, 47, 4951-4955). On the other hand, the pK_a value of the conjugate acid of NH₃ is larger than that of the conjugate acid of H₂O. Therefore, NH₃ is a poor leaving group compared to H₂O. The conclusion was indeed made based on the theory of organic chemistry and the reported literature. However, our original intention of this research is to achieve the preparation of fully conjugated benzobisthiazole-based

COFs utilizing the classical organic reaction and reported literature. Exploring the details of the reaction mechanism is not our purpose. Besides, we highly agree with the reviewer's suggestions. The new reference mentioned above has been added to the revised version. Also, we have made some revisions to more carefully describe the mechanism.

2. *In Page 5 and Figure S1, the authors mention that “The signals at about 1579 cm^{-1} could be attributed to the C=N bond stretching of thiazole rings”. However, it is a bit difficult to find similar signals at $\sim 1579 \text{ cm}^{-1}$.*

Answer: Thanks for the reviewer's question. To accurately assign the IR signals, the FT-IR spectrum (**Figure R1**) of the model compound was further measured. By comparing the FT-IR spectrum of BTH-2CN with that of the model compound, the signal at about 1553 cm^{-1} could be attributed to the C=N bond stretching of thiazole rings. Owing to the overlap and low intensity of the signals, it is often difficult to find this characteristic signal in the spectra of COFs (Nat. Commun. 2018, 9, 2600; J. Am. Chem. Soc. 2020, 142, 25, 11131-11138).

Figure R1 The FT-IR spectra of the model compound.

3. *In Figure 2. Although some assignments for PXRD have been deleted, some unknown peaks are still there. The pore size distributions are still not good. Considering there are also side peaks for aldehyde groups in solid ¹³C-NMR, it is still difficult to know whether the following good performance comes from the 2D COF structure or possible impurities.*

Answer: Thanks for the reviewer's question. This is a very good suggestion. Most of the synthesized COFs are polycrystalline. As a result, COFs often failed to show PXRD signals like single crystals. Moreover, the fully sp²-carbon conjugated COFs often could not show the signals of every crystal plane clearly due to their poor reversibility of the bond formation compared with the imine linkage. To prove the photocatalytic performance does come from the 2D COF structure, the photocatalytic performance of the monomers and model compound has been further explored. The monomers and model compound did not show any photocatalytic activity towards hydrogen evolution. Besides, as mentioned in the former response, the obtained COFs have been washed with tetrahydrofuran and ethanol via soxhlet extraction for 2 d to remove the impurities. Thus, the good photocatalytic performance could be attributed to the 2D conjugated structure. The related description has been added to the revised version.

4. *The authors still do not provide solid explanation for the high hydrogen evolution rate, and only the one sentence in summary "Three fully sp²-carbon conjugated 2D COFs containing benzobisthiazole units as photoactive group" is still superficial. In particular, the newly added information "control experiments without the existence of Pt or sacrificial electron donors were also performed. No hydrogen evolution process was observed." Also, the results cannot confirm the photoactive nature of benzobisthiazole groups.*

Answer: Thanks for the reviewer's question. As displayed in **Figure 3** and **Figure S5**, the photocurrent measurements, electrochemical impedance spectroscopy and solid-state absorption spectra have been performed to explore the possible reasons for

the high hydrogen evolution rate. These data have explained the origin mechanism for the differences in the photocatalytic performance. These measurements have been widely used to study the mechanisms. To further pry into the mechanisms, we also did the transient absorption spectroscopy of the obtained COFs in collaboration with Prof. Wu from Anhui Normal University that we have added into the Acknowledgement. The problem is the solubility of the COFs materials. We tried sonication to maximize the dispersion. Also, the spin-coating method to obtain the film has been adopted. Unfortunately, all our attempts were to no avail. No positive signals were obtained.

This research also suggests that integrating the D-A structure into the sp^2 -carbon conjugated COFs will greatly enhance the photocatalytic performance. To verify the photoactivity of benzobisthiazole group, a photocatalytic hydrogen evolution experiment of the model compound was performed. Unfortunately, no hydrogen evolution process was observed. However, the benzobisthiazole structure as an important building block has been extensively used to construct optoelectronic materials. Thus, we have made some revisions to facilitate a more objective description.

Reviewer 3:

I am satisfied with the revision and would like to recommend the publication after addressing one minor concern: The authors claimed the lower AQYs at 420 nm and 450 nm than that at 500 nm can be ascribed to the difference of absorption intensity. However, the absorption spectra could not prove this speculation. The authors can give a more reasonable explanation here.

Response: We sincerely appreciate the reviewer for the kind and objective comments. This is a very good question. In addition to the difference of absorption intensity, the energy level may be another important reason. The energy level at 500 nm is more closer to the ideal optical band gap of about 2.3 eV (*Angew. Chem. Int. Ed.* 2018, 57, 14188-14192).

REVIEWERS' COMMENTS

Reviewer #1 (Remarks to the Author):

The authors have carefully addressed my comments in the revised manuscript. I would recommend the publication of the current manuscript.

Reviewer #3 (Remarks to the Author):

I am satisfied with the revision and recommend the publication.

Reviewer 1:

The authors have carefully addressed my comments in the revised manuscript. I would recommend the publication of the current manuscript.

Response: We sincerely appreciate the reviewer for recommending the manuscript's publication.

Reviewer 3:

I am satisfied with the revision and recommend the publication.

Response: We sincerely appreciate the reviewer for recommending the manuscript's publication.